# Rapid Testing and Interventions to Control *Legionella* Proliferation following a Legionnaires’ Disease Outbreak Associated with Cooling Towers

**DOI:** 10.3390/microorganisms9030615

**Published:** 2021-03-17

**Authors:** Charlotte Young, Duncan Smith, Tim Wafer, Brian Crook

**Affiliations:** 1Health and Safety Executive, Buxton SK17 9JN, UK; charlotte.young@hse.gov.uk; 2Health and Safety Executive, Newcastle NE98 1YX, UK; duncan.smith@hse.gov.uk; 3Water Solutions Group, Harrogate HG3 1EQ, UK; tim.wafer@watersolutionsgroup.org.uk

**Keywords:** *Legionella*, cooling towers, qPCR, environmental water samples, routine monitoring, rapid detection, trend analysis, control effectiveness feedback

## Abstract

Most literature to date on the use of rapid *Legionella* tests have compared different sampling and analytical techniques, with few studies on real-world experiences using such methods. Rapid tests offer a significantly shorter feedback loop on the effectiveness of the controls. This study involved a complex of five factories, three of which had a history of *Legionella* contamination in their cooling water distribution system. Multiple sampling locations were utilised to take monthly water samples over 39 months to analyse for *Legionella* by both culture and quantitative polymerase chain reaction (qPCR). Routine monitoring gave no positive *Legionella* results by culture (*n* = 330); however, samples were frequently (68%) positive by qPCR for *Legionella* spp. (*n* = 1564). *Legionella* spp. qPCR assay was thus found to be a good indicator of cooling tower system health and suitable as a routine monitoring tool. An in-house qPCR limit of 5000 genomic units (GU)/L *Legionella* spp. was established to trigger investigation and remedial action. This approach facilitated swift remedial action to prevent *Legionella* proliferation to levels that may represent a public health risk. Cooling tower operators may have to set their own action levels for their own systems; however, in this study, 5000 GU/L was deemed appropriate and pragmatic.

## 1. Introduction

*Legionella* bacteria exist in low concentrations in natural water sources such as rivers, lakes, and reservoirs, with minimal risk of causing human infection. However, when they colonise and proliferate in water systems in the built environment, are then spread by aerosol generation, and inhaled by susceptible individuals, outbreaks of respiratory illness can occur. Hot- and cold-water systems, spa pools, and industrial sources using process water are at risk of causing potentially fatal pneumonia-like Legionnaires’ disease (LD), or the generally milder and self-limiting Pontiac and Lochgoilhead fevers [1,2,3].

Cooling, required by industrial processes to dissipate excessive heat, or for freezing/chilling, can be achieved by using cooling towers (CTs) or evaporative condensers, collectively referred to as evaporative cooling systems (ECS). Typically, these interface large airflows and sprayed water with a temperature differential, and it is likely that such systems will create an aerosol. Physical barriers on CTs to control this, commonly termed drift eliminators, will trap most, but not all aerosols. Therefore, if the cooling water is contaminated by *Legionella*, it is possible for aerosols to be dispersed over a wide area, potentially exposing workers on site, neighbouring workplaces, or nearby members of the public [4]. Factors that contribute to growth of *Legionella* bacteria in cooling water include storage and/or re-circulation of water at temperatures between 20 and 45 °C, and a source of nutrients, for example, the presence of sludge, scale, or fouling [5,6]. The bacteria can grow rapidly; therefore, as well as reducing aerosol generation and/or spread, it is important to control growth by maintaining a clean plant and process water [7,8]. This includes the correct use of validated and stable biocidal treatment; for example, chlorine is unstable in the presence of organic matter and may be rapidly neutralised [7,8,9].

Sporadic outbreaks of Legionelloses have resulted from contaminated industrial water systems and have involved ECS. Outbreaks have ranged in scale both in terms of the numbers of people infected and severity (illness and fatality) [10]. When ECS-related outbreaks occur, they can not only infect workers on site but can also, due to ECS aerosol emissions and the exit velocities involved, spread aerosols of *Legionella* bacteria several hundred metres from the point source and thus infect members of the public offsite [10,11,12,13]. This therefore constitutes a major public health concern, particularly in densely populated areas, or when contaminated ECS are located near susceptible individuals, such as those found in hospitals.

Routine monitoring for *Legionella* in CTs is required by most regulatory bodies as a check on the effectiveness of the control and precautions in place. Of the existing test methods, culture is considered the “gold standard” [14]. Alternative techniques for monitoring the presence of *Legionella* have been developed for on-site use, and quantitative polymerase chain reaction (qPCR) has been considered promising [14,15]. Being reproducible and a good indicator of microbiological water quality, qPCR is a suitable complementary method to culture for (i) routine surveillance, (ii) to monitor changes in *Legionella* concentration, and (iii) for rapid corrective action. The latter is achievable because of the rapid feedback time from qPCR (i.e., typically 1 day), which enables intervention in a water system to reduce the potential for associated infection risks. Rapid response could also potentially lead to reduced biocide costs and ecological benefits [16]. However, because qPCR measures levels of DNA, precise determination of actual bacterial numbers is difficult. At present in the UK, alert and action numerical levels are prescribed for culture-based *Legionella* detection [7], but not for qPCR-based levels. A single qPCR assay may thus be of limited value for risk monitoring, although with contextual information can be valuable for trend analysis [16,17,18,19,20]. The qPCR assay will also indicate the presence of *Legionella* species other than *Legionella pneumophila.* This means that in certain conditions it may follow trends better than culture methods designed to detect only *L. pneumophila*; however, this may depend on the specific cooling system [20]. Detecting no or low numbers of bacteria by qPCR is a strong indication for minimal risk [19]. Rapid detection of large numbers of *Legionella—*especially *Legionella pneumophila*—is valuable as an indicator of risk, although qPCR may lead to false positives when compared to culture results [19].

For *Legionella* analysis in water by culture, most laboratories use methodology described in an international standard [21]. However, culture-based analysis can take up to 14 days to obtain a result, with the results often variable, through poor recovery. More recently, an international standard method has been developed for rapid (same-day) *Legionella* qPCR for water samples [22]. Where alternative methods to culture are used, these should be validated to show they perform at least as well. Reliably detecting the presence of *Legionella* is technically difficult, whether by traditional culture methods or molecular detection, especially where the methods require specialist laboratory facilities [14,21,22,23,24].

Several commercial qPCR kits are currently available, the main differences between them being the degree of standardisation of the three critical steps: DNA extraction, qPCR preparation, and data analysis [16]. Interpreting the results from *Legionella* qPCR assay of environmental samples is difficult, as a negative result is no guarantee that *Legionella* bacteria are absent and, in addition, does not necessarily mean that the water is safe. Conversely, a high local count may not indicate an overall failure of system controls [24]. While maintaining undetectable levels of *Legionella* by culture is often the gold standard for water management, having undetectable levels of *Legionella* DNA may not be a reasonable expectation in all settings. In a significant study by the US Centers for Disease Prevention and Control (CDC), *Legionella* DNA was found to be present in the majority of CTs sampled from across the USA. Consequently, even with an effective water maintenance programme, *Legionella* DNA from killed or inhibited bacteria can be expected to be present in CT water [25].

The performance of both qPCR and culture techniques are influenced by several factors. Firstly, the characteristics of the water matrix (e.g., background microorganisms, inhibitory substances) can affect both, but with the sensitivity of qPCR affected by water quality this can make it difficult to compare and interpret the results by qPCR and culture in all samples [20]. Greater discrepancies in results are observed in dirtier water samples, which is of concern, as these are more likely to be susceptible to *Legionella* colonisation [14]. Secondly, differences could occur between the limits of quantification with culture and those with qPCR due to (i) the presence of suspended solids and inhibitors in cooling waters, (ii) the difference in volumes of water filtered for analysis, (iii) the different dilution factors being applied, and (iv) different filtration–extraction procedures [20].

The lack of a direct correlation between culture and qPCR does not necessarily mean that culture is the more reliable or the most appropriate method for protecting public health. In fact, Lee, et al. (2011) suggested that, in future, culture may not necessarily be considered the gold standard [26]. However, as qPCR tends to yield consistently higher genomic units (GU)/L concentrations than colony-forming unit (cfu)/L concentrations by culture [20], samples were 50–100% more likely to return a positive result by qPCR than by culture when analysed concurrently [25,27]. Markedly higher concentrations of both *Legionella* species and *L. pneumophila* are measured by qPCR rather than by culture [15,25]. However, the difference in positivity rates between culture and qPCR in complex water samples has been considered more likely to be due to false-negative culture results, rather than to false-positive results by the qPCR method [11]. A study by Collins, et al. (2017) showed that although there was only a weak correlation between the results of CT water samples by both qPCR and culture, they did follow similar trends [28].

Culture enumeration can underestimate the risk of *Legionella* due to, among other issues, (i) inability to count viable but non-culturable (VBNC) organisms, (ii) the slow growth rate of *Legionella* on agar media, (iii) overgrowth by accompanying organisms, (iv) presence of vesicles containing *Legionella* expelled from protozoa that prevent culture, or (v) loss of cultivability during sample holding time prior to culturing [16,28]. In addition, the GVPC agar medium used in standard analysis is highly selective for *Legionella pneumophila*, thus making quantitative comparison of *Legionella* species by culture and qPCR difficult [20,26].

The qPCR assay does have limitations. The detection of both living and dead bacteria raises concerns about false-positive results in water systems due to contamination with residual dead cells or free DNA, thus complicating any evaluation of the real health risk. Moreover, samples may show complete or partial qPCR inhibition. This averaged 9.8% of CT samples across five studies, due to presence of humic acids and other inhibitory compounds in complex water systems, leading to false-negative results [14,16,17,18,25,29]. The negative predictive value (NPV) of the *Legionella* qPCR assay, however, is normally very high, and therefore failure to detect is (i) a strong indication that the risks from *Legionella* within the CT system are under control, (ii) a useful negative screening tool to rule out potential sources in an outbreak situation, and (iii) a useful indicator for the restarting of a system implicated as the source of an outbreak following cleaning and disinfection [16,19,26,28,30].

Most peer-reviewed papers to date that relate to the use of qPCR for ECS monitoring have been cross-sectional studies to compare qPCR with culture. Few papers detail longitudinal studies to show the use of qPCR for trend analysis and routine monitoring at the same locations over time. One study was found that did monitor the same *L. pneumophila-*contaminated CT for 13 months by analysing 104 serial samples. The culture and qPCR results were reported as being extremely variable over time; however, the curves were similar. The differences between the qPCR and culture results did not seem to change over time and were not affected by regular biocide treatment of the CT. The authors concluded that the qPCR assay for *L. pneumophila* could permit more timely disinfection of CTs [16].

Our paper describes a longitudinal study over a 39-month period of CTs previously identified as the source of an outbreak of LD. It also provides some real-world user experience of the efficacy of qPCR being deployed as a novel rapid monitoring tool to assess the effectiveness of controls.

An LD outbreak in 2015 was traced to a factory complex in northeast England and, as part of the remediation works, the dutyholder instigated enhanced *Legionella* monitoring by installing many more sampling points on the cooling water distribution system and implementing a comprehensive programme of monitoring involving culture and qPCR. An in-house action limit for qPCR levels within the water system was established on the basis of initial trend analysis of the results. The *Legionella* species qPCR results were deemed to be a good indicator of system health, and any elevated levels enabled rapid investigation and remediation before any sub-optimal control of *Legionella* proliferation could develop into a situation increasing risk in both occupational and public health contexts. This paper examined the relationship between the qPCR data, interventions, and the effectiveness of remedial actions.

## 2. Materials and Methods

The study site comprised a complex of 5 factories on a business park, referred to as Factory 1, Factory 2, Factory 3, Factory 5, and Factory 8 (Figure 1). Four CTs (CT1, CT3, CT5, and CT6) were co-located, as also shown in Figure 1. Factories 1, 2, and 5 were where the production activities required cooling water. In Figure 2, Figure 3 and Figure 4, schematic plans of Factories 1, 2, and 5 respectively show the location of sampling points on the cooling water distribution pipework. For simplicity, pipe runs and individual machines were not included in the schematics in Figure 2, Figure 3 and Figure 4, but in terms of pipework layout, Factory 5 was the least complicated, then Factory 2, and with Factory 1 having the most machines, most complicated pipework, and as a consequence the greatest number of sampling points. Factories 2 and 5 shared 2 CTs which have linked sumps and are effectively considered as a single CT (referred to as CT3/6). This is located on one side of Factory 2, with flow and return supply pipework to both factories. Factory 1 was served by 2 CTs (CT1 and CT5) located away from each other on 2 sides of the factory. In total, there was reported to be over 10 km of cooling water distribution pipework within the 3 factories, serving numerous machines such as extruders, co-extruders, injection moulders, and other similar production machines.

*Legionella* contamination was not confined to 1 factory, or 1 specific CT, and affected workers during the outbreak were located in all 3 factories, with workers routinely moving between factories. Figure 1 also shows main roadways between the factories. Road and/or pedestrian access ran close by to CT1 and CT3/6, and in addition, 2 designated smoking areas, as shown in Figure 1, were used by some workers, 1 of these being outside Factory 5 opposite CT3/6.

In total, 60 sampling locations were identified around the cooling water distribution pipework that corresponded with different production lines and reflecting their complexity: 34 sample points in Factory 1, 19 sample points in Factory 2, and 7 sample points in Factory 5. Where a production line had more than 1 sampling location (i.e., B57A and B57B), for the purposes of this study, the results were combined and reported as 1 sampling location (i.e., B57). The strategy was to conduct sampling on a monthly basis. In practice, not every production line was in operation at every round of sampling, and thus mostly it worked out that 50% of locations were sampled on 1 month with the other 50% sampled the following month.

Following standard sampling protocols [31], suitably trained environmental consultants collected up to 1L water samples in sterile containers with sodium thiosulfate to neutralise any biocide residue. These were transported in dark conditions at room temperature to an accredited *Legionella* testing laboratory (Intertek ITS Testing; UKAS 4065) for processing within 24 h of collection.

General microbiological parameters were tested, including aerobic colony count (data not presented). *Legionella* analyses comprised culture-based analysis following standard protocols [21], and analysis by qPCR for both *Legionella* species and *Legionella pneumophila* [22]. For qPCR, analysis was performed using a Biorad CFX System with 2 different Biorad IQ Check *Legionella* test kits, 1 for *Legionella* species and 1 for *L pneumophila*. DNA extraction prior to analysis was done with Aquadien Extraction Kits. Data were presented for culture-based analysis as cfu/L with a lower limit of detection (LOD) of 50 cfu/L. For qPCR, data were presented as GU/L, where detected, for both *Legionella* species and *L pneumophila*, as “positive but below the LOQ” (limit of quantification) if close to the LOQ of 1000 GU/L, or “negative” if below the LOD.

The factory owners and maintenance staff (referred to collectively as the CT operators), in agreement with the environmental consultants, set an in-house action limit of >5000 GU/L *Legionella* spp. detected by qPCR. This value was considered to be one below which *Legionella* levels in the cooling water system could be considered under control, but above which could be considered at risk of significant proliferation and warranted actions such as adjusting dose rate of the hypochlorite-based biocide routinely delivered into the cooling water circuit. A value of >100,000 GU/L) *Legionella* spp. in a sample was considered to be very high and triggered interventions to reduce colonisation levels. These remedial interventions included a local investigation of an area where the positive result was found, for example, to check for interruptions to water flow or a local source of potential contamination. Remedial actions taken included turning on a chlorine dioxide generator to supplement the routine biocide dose.

Sampling data were provided to the authors from January 2017 to March 2020, although no culture-based sampling results were provided for 2020. These data were collated, cleaned, and interrogated.

## 3. Results

### 3.1. Culture

A total of 330 *Legionella* spp. culture sample results were reported for the period of January 2017 to December 2019. Sampling was undertaken on a monthly frequency, initially at all 60 sampling locations (pipework and CTs) across the three factories for January–March 2017. This was then reduced to the CTs (CT1, CT3/6, and CT5) only from April 2017 onwards as the analysis cost and a lack of positive culture results did not justify the resources to continue this level of sampling.

All of these 330 sample results were negative by culture (i.e., below the LOQ of 50 cfu/L). The negative sampling results did not allow any direct comparisons to be made between culture and qPCR results, other than to say that the various contamination events that did take place would not have been detected by the traditional “gold standard” monitoring method.

### 3.2. qPCR

#### 3.2.1. *Legionella* spp. qPCR Results

In total, 1564 sample results were obtained between January 2017 and March 2020, with 505 samples obtained in 2017, 460 in 2018, 464 in 2019, and 135 in 2020.

From the 60 sampling locations across the three factories, where samples were taken monthly, the number of samples taken at each sampling location over the time period ranged from 1 to 116 (sometimes multiple samples were taken at one location—up to a maximum of seven).

Overall, 496 (32%) of the *Legionella* spp. sample results were negative and 1068 (68%) were positive. Of those positive, 848 (54% of the total number) were positive but <LOQ, 114 (7%) were positive but <5000 GU/L *Legionella* spp., and 106 (7%) were positive with values >5000 GU/L *Legionella* spp. Over the sampling period, 46 (73%) of the sampling locations had at least one positive (above LOQ) result, and in total 35 (55%) of these sampling locations had at least one result >5000 GU/L (the in-house action limit). The smallest positive (above LOQ) result was 1307 GU/L and the largest was 476,000 GU/L.

Table 1 summarises data for the sample points for which values greater than the in-house action limit (>5000 GU/L *Legionella* spp.) were recorded in more than five samples over the study period, with these being nine sample points, all from Factory 2. Figure 5 shows *Legionella* spp. GU/L data for the nine sample points over the three-year sampling period.

Risk assessments of cooling water systems conventionally indicate that CTs would be a high-risk system and therefore a target for monitoring. However, as shown in Table 1 and Table 2, sampling from the CT itself, may not be the best monitoring location and may not represent the “worst case”. Similar numbers of samples were taken from each (40 from CT1, 42 from CT5, and 39 from CT3/6), with no samples having values >5000 GU/L *Legionella* spp. from CT5, and only two samples (5% of the total) having values >5000 GU/L *Legionella* spp. from CT1 and four samples (10%) from CT3/6. Figure 6 shows *Legionella* spp. GU/L data for the CT sample points over the three-year sampling period.

Samples recording very high (>100,000 GU/L) *Legionella* spp. triggered interventions to reduce colonisation levels. Although many dutyholders intervene to rectify *Legionella* problems by superchlorination, in this instance, the factory maintenance regime deemed most appropriate was to use, for a period of a few days, a back-up online chlorine dioxide generator installed specifically for the purpose. This was followed by enhanced monitoring. Table 3 summarises all results >100,000 GU/L, their location, and the date on which they recorded the high value. Additionally included are the previous results obtained, which in some instances did not show quantifiable qPCR values predicting potential contamination. In all instances, samples taken after and within 8 days of a high recorded value, and after remedial intervention, showed significant reductions in numbers to below 5000 GU/L.

#### 3.2.2. Legionella pneumophila qPCR Results

For *Legionella pneumophila* detection by qPCR, 72 (4.6% of the 1564 sample results) samples were positive, of which only 8 (0.51%) samples were above the LOQ and 3 were >5000 GU/L.

Seven of the eight samples that were greater than the LOQ occurred in July 2018, and the results are shown in Table 4 below.

Where there was a qPCR *Legionella pneumophila* spike, there was also a qPCR *Legionella* species spike, and the magnitude of the *Legionella* spp. results was greater than the *Legionella pneumophila* results.

#### 3.2.3. July 2018 Results Spike

In July 2018, there was a noticeable spike in both the magnitude of the results, and also the number of results testing positive. All the CT water sample results (36 out of 36) in July 2018 tested positive by the *Legionella* spp. qPCR assay. Of these samples, 19 (53%) *Legionella* spp. results were above the LOQ (i.e., >1000 GU/L) and 8 (22%) were >100,000 GU/L.

In addition, 30 (83%) *Legionella pneumophila* sample results tested positive, with 8 (22%) being greater than the LOQ (1000 GU/L) and 2 (5%) being >100,000 GU/L.

Subsequent site investigation found problems with the cooling water circulation system in Factory 2 (and therefore water stagnation and a consequent lack of circulation of biocidally treated water) resulting from an isolation valve closure necessitated by a production machinery move and the inadvertent failure to reopen the valve upon completion of the work.

A water sample from cooling tower CT3/6 was analysed by culture and found to be negative. This sample was collected on the same day in July 2018 as the samples reported above for qPCR analysis. Quarterly culture sampling and analysis (the recommended minimum sampling frequency and monitoring method for ECS in Great Britain) may not have detected this particular event for some time, by which time significant *Legionella* proliferation could have occurred [23].

#### 3.2.4. March 2020 Results Spike

In March 2020, there was a second noticeable spike in the results. There were 57 samples taken, of which 41 (71.9%) were positive. Seventeen samples (29.8%) were above the LOQ and, of these, 14 (24.8%) were >5000 GU/L. All the positive results above the LOQ were from Factory 2.

Site investigations found new pipework dead-legs in Factory 2, which were created after the move of production machinery to a new factory site.

## 4. Discussion

It is important that the data from alternative test methods to culture can be properly interpreted so that appropriate alert or action levels can be set to enable informed decisions on the control measures needed. This may be achieved by running the tests in parallel with traditional culture-based methods for a period [23].

In this study, none of the culture-based analyses (*n* = 330) were positive for *Legionella* spp., i.e., all were below the LOQ of 50 cfu/L. This included periods where very high levels were detected by qPCR. This may have been a consequence of various factors that could compromise culture including interference of the assay, competition from other bacteria present, and the selectivity of the agar isolation medium used [14,20,26,28]. In the UK, it has been noted that only about 5% of CT water samples test positive by culture (unpublished meta data, shared with the authors, from a commercial testing laboratory).

The use of qPCR for the detection of *Legionella* has posed difficulties for interpreting results in the context of quantification of health risk and demonstration of the effectiveness of the applied controls and precautions. Regulations and guidance typically specify target levels on the basis of culture in terms of cfu/L, while those of qPCR are expressed as GU/L. This difference has sometimes led to confusion as these results are not interchangeable for a variety of reasons [24].

As also highlighted in previous studies, using the qPCR result for *Legionella* samples may overestimate the risk of infection [20,25,27]. This may be further complicated by qPCR assay detecting bacteria that have entered the VBNC state, therefore detectable but not capable of causing infection, although differences between PCR and culture were not more marked during periods of decontamination [16]. However, the rapid detection by qPCR of high concentrations of *Legionella*—especially *Legionella pneumophila*—is valuable as an indicator of risk, although it may be false positive compared to culture results [14]. Caution must be exercised to avoid unnecessary and expensive emergency decontamination procedures [19,32].

Previous studies have concluded that whilst it is tempting to suggest a potential for threshold values for qPCR that quantify an abundance of *Legionella* DNA at which corrective action would be required, correlations between the qPCR crossing threshold values for detection of *Legionella* DNA and the ability to culture *Legionella* bacteria were not found [25].

The qPCR assay is undoubtedly useful as a complementary tool for the rapid routine monitoring of *Legionella* trends at CT sites; however, it is important that data from such tests can be properly interpreted in order to enable informed decisions on the effectiveness of control measures. The results of positive qPCR samples are difficult to interpret, as the assay detects *Legionella* DNA from both live and dead bacterial cells and the units of measurement are not directly comparable with published action and alert levels for culture (expressed in cfu/L).

Four publications have proposed qPCR action and alert levels for CTs, which are shown in Table 5 below.

Three of these publications proposed action and alert levels for the *Legionella pneumophila* qPCR assay, as this species is directly associated with public health risk [26,33,34]. In addition, two publications proposed alert and action levels for *Legionella* spp., despite there being less association with the genus and public health risks from CTs [26,28].

However, if the environmental conditions within the cooling water do support *Legionella* spp., then they would also support the proliferation of *Legionella pneumophila*, if present. The French Agency for Food, Environmental and Occupational Health and Safety (ANSES) postulated that monitoring of *Legionella* spp. may benefit facility operators, particularly in terms of detecting malfunctions or limiting drift in concentrations of microorganisms, especially those related to *Legionellae*, other than *Legionella pneumophila*. ANSES advised caution, however, before potentially replacing culture action and alert levels, as there should be an evaluation of the adequacy of monitoring and control procedures for microbial proliferation in CTs. They suggest a relative increase of 2 log or higher in the concentration of *Legionella* spp. in the installation system water compared to the makeup water could be an indicator of a lack of CT control effectiveness [33]. In our dataset, only one qPCR result from the mains makeup water was reported (date: 17 July 2018, result: positive, but <LOQ), and therefore this comparator could not be applied in this instance.

Collins et al. (2017) proposed qPCR action and alert levels for CTs. However, these were not derived from actual CT data. In their paper, they advised that the detection of *Legionella* spp. by qPCR at >1000 GU/L should warrant further investigation in CTs, given their potential public health risk. It should be noted though that the qPCR LOQ for some analyses may actually be higher than 1000 GU/L, particularly if the sample required dilution, effectively rendering the suggested alert level lower than may be practical for the interpretation of some CT water results [28].

In the case study presented here, the CT operator was able to establish what was considered “typical” for the CTs on site and subsequently set an in-house limit of 5000 GU/L for *Legionella* spp. to use for routine monitoring purposes. This value was seen to be a good indicator of system health, and the site operators were able to work to this value, as both a combined alert and action limit. Any results in excess of this value prompted action, i.e., local investigation of water flow and contamination in the area of the elevated result; addition of chlorine dioxide to the system; as well as checking and adjustment, if required, of the on-demand online hypochlorite-based biocide dosing system, etc. Whenever elevated levels of *Legionella* spp. were reported from a specific sampling location on the cooling water distribution system, a local investigation in that area invariably found the source of the problem, and site staff were able to remedy the situation before it could develop into an increased risk to public health. This adoption of a pragmatic real-world action level and shortening the feedback loop between the collection of a water sample and the undertaking of corrective action was invaluable. Since the adoption of this intervention, there have been no further cases of LD associated with this cooling water distribution system.

## 5. Conclusions

As routine *Legionella* monitoring is generally undertaken to determine the effectiveness of the controls and precautions, a rapid feedback on this effectiveness is highly desirable. The qPCR assay for *Legionella* spp. has been shown to be a good indicator of CT system health, and spikes in results are likely to indicate system changes and/or sub-optimal control and enable much speedier remediation.

CT operators may use trends from the results from their own systems to determine what is normal for them when these are under control and establish their own in-house limits accordingly. In this study, 5000 GU/L for *Legionella* spp. was shown to be a suitable limit and also to be a highly indicative predictor of a negative culture result. This limit is quite conservative when compared to other proposed qPCR limits, but it is based on real-world experiences and may also be suitable for other CT operators.

## Figures and Tables

**Figure 1 microorganisms-09-00615-f001:**
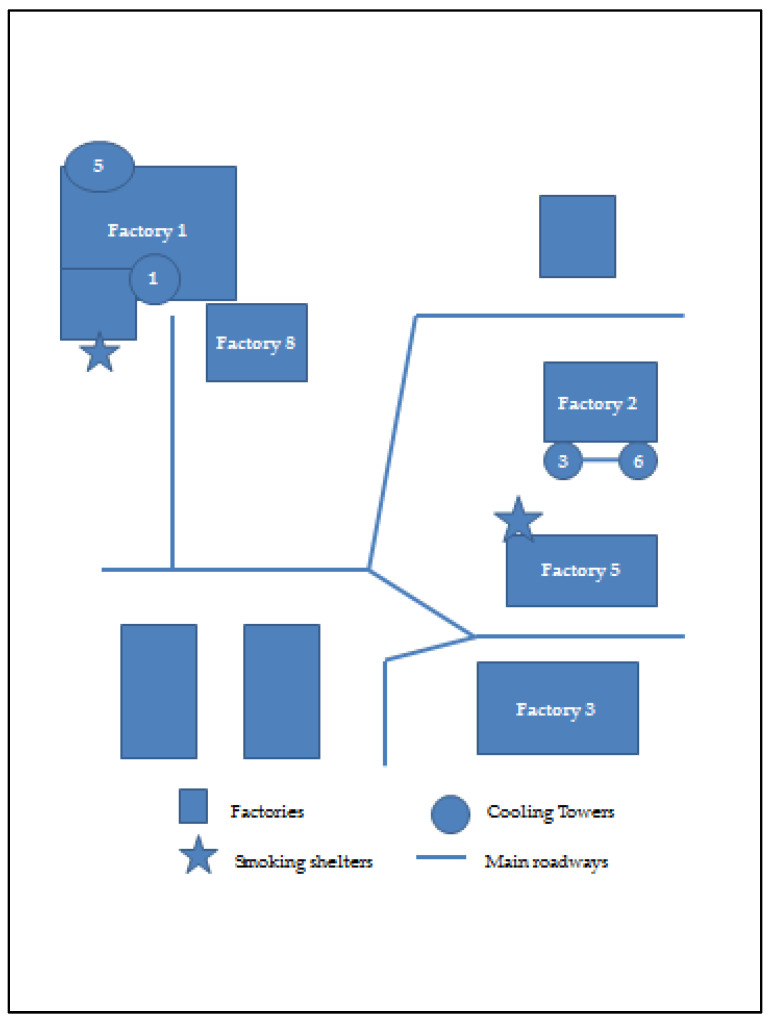
Layout of factory complex showing cooling towers (CTs), smoking shelters, and main roadways.

**Figure 2 microorganisms-09-00615-f002:**
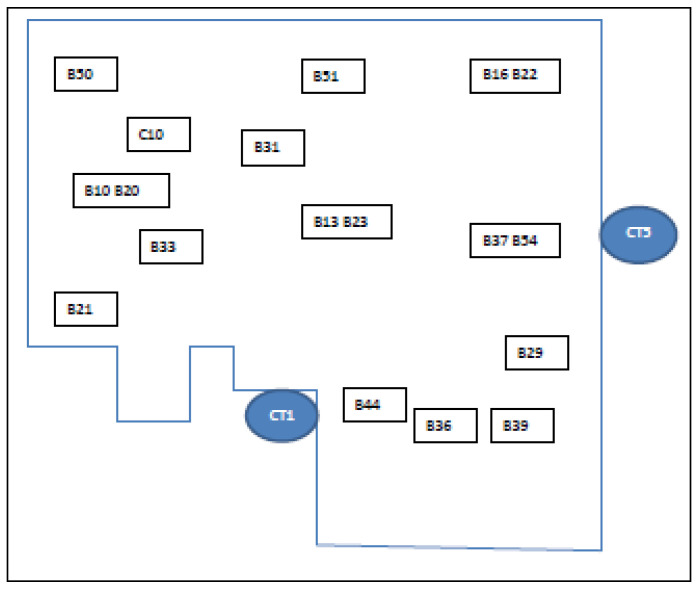
Schematic of Factory 1, showing most of sample points. CT1 and CT5 = cooling towers where samples were also taken.

**Figure 3 microorganisms-09-00615-f003:**
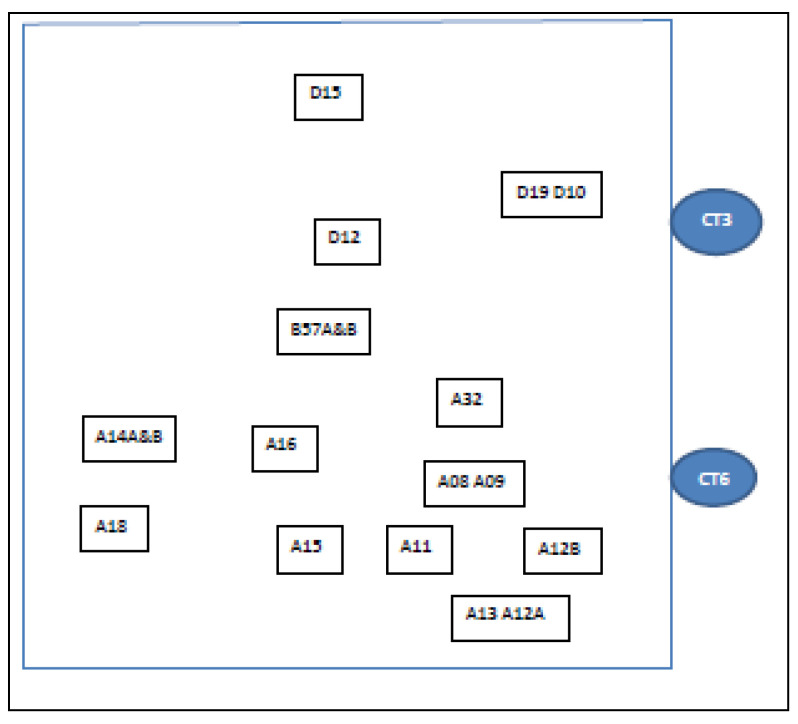
Schematic of Factory 2 showing most of locations of drain/sample points. CT3/CT6 = two cooling towers (with connecting pipework, so treated as one) where samples were also taken.

**Figure 4 microorganisms-09-00615-f004:**
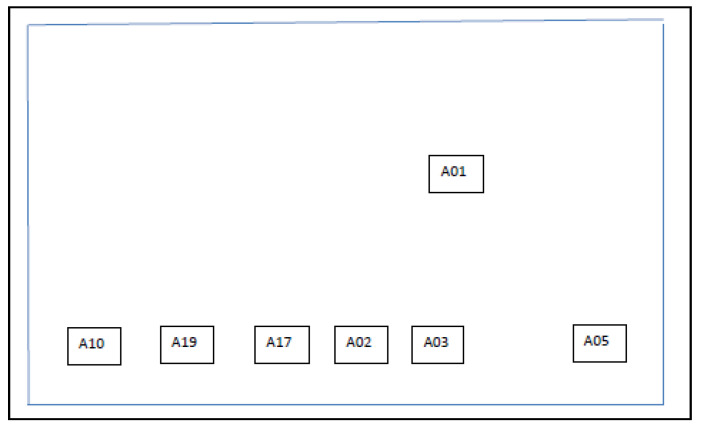
Schematic of Factory 5 showing locations of drain/sample points fed by CT3/6 from Factory 2.

**Figure 5 microorganisms-09-00615-f005:**
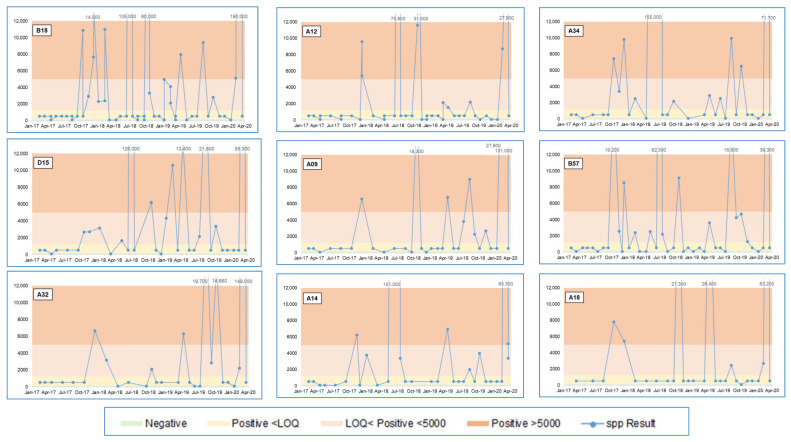
*Legionella* spp. qPCR data over three years for nine locations in Factory 2 with the greatest number of >5000 GU/L results.

**Figure 6 microorganisms-09-00615-f006:**
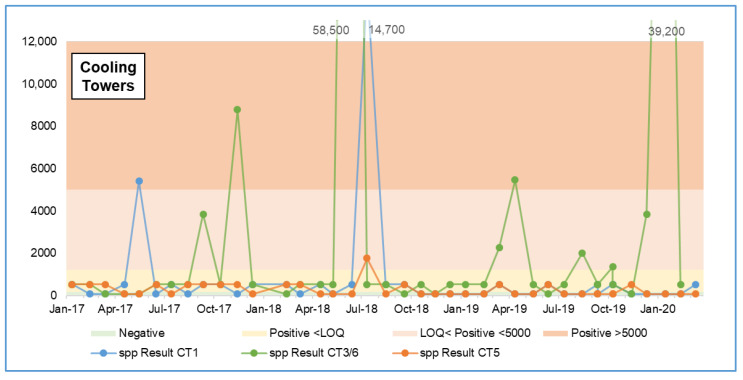
*Legionella* spp. qPCR data for the cooling towers over the three-year period.

**Table 1 microorganisms-09-00615-t001:** Sample points where greater than five samples recorded >5000 genomic units (GU)/L *Legionella* spp.

Sample Location	Factory	Negative		Positive <LOQ		Positive <5000		Positive >5000		Total
		N	%	N	%	N	%	N	%	
B18	Factory 2	11	19%	29	50%	8	14%	10	17%	58
A12	Factory 2	12	24%	29	57%	3	6%	7	14%	51
A34	Factory 2	5	16%	15	48%	5	16%	6	19%	31
D15	Factory 2	3	9%	17	52%	7	21%	6	18%	33
A09	Factory 2	4	13%	18	58%	3	10%	6	19%	31
B57	Factory 2	13	28%	19	41%	8	17%	6	13%	46
A32	Factory 2	5	18%	14	50%	4	14%	5	18%	28
A14	Factory 2	7	21%	17	50%	5	15%	5	15%	34
A18	Factory 2	1	4%	20	71%	2	7%	5	18%	28

**Table 2 microorganisms-09-00615-t002:** *Legionella* spp. qPCR data for cooling towers.

Sample Location	Factory	Negative		Positive <LOQ		Positive <5000		Positive >5000		Total
		N	%	N	%	N	%	N	%	
CT1	Factory 1	21	53%	17	43%	-	0%	2	5%	40
CT5	Factory 1	27	64%	14	33%	1	2%	-	0%	42
CT3/6	Factory 2	8	21%	22	56%	5	13%	4	10%	39

**Table 3 microorganisms-09-00615-t003:** Areas with readings >100,000 and values for the samples taken before and after.

Sample Location	Test Month before	Result before	Month High	High Result	Test Month after	Result after
A01	9 May 2018	Positive <LOQ	11 July 2018	258,000	17 July 2018	1433
A08	9 May 2018	Positive <LOQ	11 July 2018	113,000	17 July 2018	Positive <LOQ
A08	12 February 2020	5715	11 March 2020	476,000	19 March 2020	Positive <LOQ
A09	12 February 2020	27,800	11 March 2020	131,000	19 March 2020	Positive <LOQ
A14	9 May 2018	Positive <LOQ	11 July 2018	151,000	17 July 2018	3374
A15	14 August 2017	Negative	13 October 2018	219,000	11 December 2017	1918
A16	9 May 2018	Positive <LOQ	11 July 2018	113,000	17 July 2018	Positive <LOQ
A32	12 February 2020	2208	11 March 2020	149,000	19 March 2020	Positive <LOQ
A34	14 April 2018	Negative	11 July 2018	155,000	17 July 2018	1148
B18	13 June 2018	Positive <LOQ	11 July 2018	105,000	17 July 2018	1185
B18	12 February 2020	5114	11 March 2020	190,000	19 March 2020	Positive <LOQ
D12	13 June 2018	Positive <LOQ	11 July 2018	139,000	17 July 2018	Positive <LOQ
D15	13 June 2018	Positive <LOQ	11 July 2018	126,000	17 July 2018	Positive <LOQ

**Table 4 microorganisms-09-00615-t004:** Areas with *Legionella pneumophila* qPCR readings >LOQ (and comparison to *Legionella* spp. qPCR readings).

Test Date	Factory	Sample Location	*Legionella* spp. Result (GU/L)	*Legionella pneumophila* Result (GU/L)
14 September 2017	1	B22	9427	3614
11 July 2018	5	A01	258,000	142,700
11 July 2018	2	A14	151,000	4021
11 July 2018	2	A34	155,000	11,790
11 July 2018	1	B31	11,300	6962
11 July 2018	1	B37	2819	1738
11 July 2018	1	B62	5701	4188
11 July 2018	1	CT1	14,700	1780

**Table 5 microorganisms-09-00615-t005:** Table to show proposed cooling tower *Legionella* qPCR action and alert levels from published literature.

Source	Parameter	Alert Level (GU/L)	Action Level (GU/L)
Lee et al. (2011) [26]	*Legionella pneumophila*	5 × 10^3^	5 × 10^4^
	*Legionella* spp.	1 × 10^5^	1 × 10^6^
ANSES [33]	*Legionella pneumophila*	5 × 10^3^	5 × 10^5^
	*Legionella* spp.	-	-
Collins et al. (2017) [28]	*Legionella pneumophila*	-	-
	*Legionella* spp.	1 × 10^3^	1 × 10^4^
PWGSC [34]	*Legionella pneumophila*	1 × 10^4^	1 × 10^5^
	*Legionella* spp.	-	-

## Data Availability

The data presented in this study are available on request from the corresponding author. The data are not publicly available due to commercial sensitivities.

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
