# Peer review of "Rapid Testing and Interventions to Control Legionella Proliferation following a Legionnaires’ Disease Outbreak Associated with Cooling Towers"

_microorganisms, 2021, doi:10.3390/microorganisms9030615_

Round 1

Reviewer 1 Report

This article discussed rapid testing to control Legionella proliferation after a Legionnaires’ disease outbreak linked to the cooling towers. Below are some of my comments for the authors:

  • Some of the important claims throughout the whole manuscript lack proper citations. Please cite appropriate sources.
  • “qPCR assay results in excess of the in-house limit of 5,000 GU/L triggered…..” How was the limit set at this value? Is there any justification? The authors described in the discussion section about various limits proposed by previous studies but 5000 GU/L was mostly proposed for Legionella pneumophila, not for Legionella spp..
  • “it is important to control growth by maintaining a clean plant and process water, and the correct use of validated biocidal treatment.” This sounds like a vague statement. Please discuss briefly various cleaning, maintenance, and disinfection techniques with specific limits/examples.
  • “and thus infect members of the public offsite….” Cite a source.
  • Italicize Legionella throughout the whole manuscript. This mistake happened multiple times.
  • “Rapid response could also potentially lead to reduced biocide costs and ecological benefits….” Is there any previous study on this?
  • “The study site comprised a complex of five factories termed Factory 1, Factory 2, Fac-tory 3, Factory 5 and Factory 8, with……” why not 1,2,3,4,5?
  • Use better quality images for the figures.
  • “with three CTs (CT1, CT3/6, and CT5)” why does the figure represent 4 CTs then? I understand that the authors described CT3/6 later on in the manuscript but make changes to the figure/text to clarify for readers.
  • “this was then reduced to the CTs (CT1, CT3/6 & CT5) only from April 2017 onwards…..” Why was that?
  • “A total of 330 Legionella spp” Number of samples should be more than that considering the description in the method section (at least 30 per month for 3 years). If that’s not the case, describe why?
  • “Sampling data from January 2017 to March 2020” this information should be moved to the Materials and Methods section.
  • “505 in 2017, 460 in 2018, 464 in 2019 and 135 in 2020.” Why culture samples were not collected in 2020?
  • “Samples recording very high (>100,000 GU/L) Legionella spp. triggered interventions to reduce colonization levels.” How did the authors set this limit?
  • “In all instances, samples taken after and within 8 days of a high recorded value, and after remedial intervention” what are the remedial interventions?
  • “Of these samples, 19 (53%) Legionella spp. results were above the LOQ…..” LOQ is mentioned as 1000, not 10000.
  • “In addition, 30 (83%) Legionella pneumophila sample results tested positive, with 8 (22%) being greater than the LOQ (1 x 104 GU/L) and 2 (5%) being >1 x 105 GU/L.” Same as above.
  • The title of the article should be changed considering the focus of the article only on testing, not on interventions.
  • “Using the qPCR result for Legionella samples may overestimate the risk of infection” describe briefly findings from previously published literature.
  • “is valuable as an indicator of risk, although it may be false positive compared to culture results.” Same as above.
  • The Discussion section did not mention anything about the VBNC state being a possible reason for higher positive results using the qPCR method.
  • What could be the possible reasons for all the 330 culture samples being negative? This is surprising to see. Did the authors collect any culture samples during July 2018 and March 2020?
  • Nothing about culture results was found in the discussion section.
  • Please revise the method and Discussion section for future submission.

Reviewer 2 Report

The manuscript by Young et al. “Rapid testing and intervention to control Legionella proliferation following a Legionnaires’ disease outbreak associated with cooling towers” reports the monitoring of three cooling towers in a factory system in 39 months after the occurrence of an outbreak.

They state that qPCR could become a useful tool to establish if a CT is safe or not.

However, several points are not so clear, in particular when and what made the “CT operator” sure that a level >5000GU/L was a risk level, so that other CT operators could do the same with other CT systems?

In several part the manuscript is difficult to read, for very long sentences.

In the Abstract, it could be made clearer where the background finishes and the work made starts, as well the conclusions the authors found.

The “Introduction” is a little bit too long and in some parts difficult to understand, especially because of too long sentences.

Aim of the study was to determine if the control measures applied to CTs, source of a previous outbreak, have been efficacious using qPCR as monitoring tool.

Results:

Clarify the period of the study and why: there is a discrepancy between culture and qPCR.

Considering the highest values of GU/L found, it is very strange that culture always was negative. A comment should be made.

Please clarify how many days after disinfection the sampling were made, considering that qPCR detects both live and dead bacteria, the high level of GU could be due to this.

Please add some notes to tables (what does it indicate “Area”) and make the legend to the figure more explicative and indicate that on y axe you report GU.

Discussion.

The authors should comment their results.

Has been it made a correlation with disinfecting practice?

Has been the incidence of cases considered? There is no other mention of the outbreak associated with the CTs a part the headline and at the end of introduction

What did determine the action level >5000 GU/L?

is there a limit for this study?

Who is the CT operator?

Author Response

Please see attachment which includes response to reviewer 1 as they cross reference

Round 2

Reviewer 1 Report

The authors have addressed my comments appropriately. I have no additional comments.